# Impact of COVID-19 on New – onset Type 1 diabetes mellitus: A six-year retrospective review from two Paediatric clinics, Kampala, Uganda

**Rosemary Ahabwenki**[1]*, **Thereza Piloya**[1], **Flavia Namiiro**[1], **Wani Muzeyi**[1],
**Catherine Nyangabyaki**[2], **Thomas Katairo**[3‡], **Aidah Namugumya**[1‡], **Joan Nakimera**[1‡]
**Fred Kyekulidde**[1‡], **Robert Kidega**[1‡], **Fozia Nizar Ali**[1‡], **Angella Mirembe**[1‡],
**Bernard Kikaire**[1], **Grace Ndeezi**[1], **Victor Musiime**[1], **Sarah Kiguli**[1]

**1** Department of pediatrics and child health, school of medicine, college of health sciences, Makerere university, Kampala, Uganda, **2** Department of Pediatrics and child health, St. Francis Hospital Nsambya, Kampala, Uganda, **3** Infectious Diseases Research Collaboration (IDRC), Kampala, Uganda

☙ These authors contributed equally to this work.
‡ TK, AN, JN, FK, RK, FNA, and AM also contributed equally to this work.
* rosieahabs@gmail.com

## Abstract

### Background

Type 1 Diabetes Mellitus is one of the most common chronic diseases in children, yet data from low-income countries remains scarce. Recent reports suggest a rising global incidence of new-onset T1DM, projected to double by 2045. The COVID-19 pandemic is believed to have further influenced T1DM onset and severity at presentation, with several studies reporting increased incidence and higher rates of diabetic ketoacidosis (DKA). This study aimed to assess the impact of COVID-19 pandemic on paediatric type 1 diabetes mellitus onset.

### Methods

We conducted a retrospective review of medical records for children aged 6 months to 19 years diagnosed with new-onset T1DM between March 2017 and February 2023 at Mulago and Nsambya hospital's paediatric diabetes clinics. Clinical and demographic characteristics were compared between the pre-COVID- 19 (March 2017–February 2020) and COVID-19 (March 2020–February 2023) periods using Chi-square and Fisher's exact tests. Trends in new onset type 1 diabetes were analyzed using interrupted time series with 12 six-month intervals. STATA 17.0 was used for the analysis.

### Results

A total of 245 children were included. The mean age at diagnosis was 10.9 years (SD ± 4.5), with 56% being female. More cases were diagnosed during the COVID-19

**Data availability statement:** All relevant data are within the paper and its supporting Information files.

**Funding:** This work was supported by the Fogarty International Center of the National Institutes of Health, U.S Department of State's Office of the U.S Global AIDS Coordinator and Health Diplomacy (S/GAC), and Presidents Emergency Plan for AIDS' Relief (PEPFAR) under award number 1R25TW011213. The content is solely the responsibility of the authors and does not necessarily represent the official views of the National Institutes of Health. The funders had no role in study design, data collection and analysis, decision to publish, or preparation of the manuscript.

**Competing interests:** The authors have declared that no competing interests exist.

period (60.4%, n = 148) compared to the pre-COVID-19 period (39.6%, n = 97), representing a 52% increase, although it was not statistically significant (p = 0.1997). Overall, 66% (n = 162) presented with DKA at diagnosis, with similar proportions in both periods (pre-COVID: 61%, COVID-19: 69%).

## Conclusion

There was a rising trend in T1DM among Ugandan children at Mulago and Nsambya pediatric diabetic clinics, from 2.7 new cases/month during pre-COVID to 4.1/month during the pandemic.

---

## Introduction

Type 1 diabetes mellitus (T1DM) is an autoimmune disorder characterized by the destruction of pancreatic beta cells, resulting in insulin deficiency and hyperglycemia [1]. It is one of the leading causes of morbidity and mortality among children and adolescents globally [1]. According to the International Diabetes Federation (IDF, 2021), over 1.2 million children and adolescents (0–19 years) are living with T1DM worldwide; however, data from low- and middle-income countries (LMICs) remains scarce [2].

Recent reports indicate a rising global incidence of new-onset T1DM in children, with projections suggesting that incidence could double by 2045 [2]. In LMICs like Uganda, children with T1DM often present with diabetic ketoacidosis (DKA) at first diagnosis, a complication associated with delayed diagnosis and poor outcomes [3,4]. Although few studies exist, available evidence suggests a growing burden of pediatric T1DM in Uganda. For instance, Bahendeka et al. (2019) reported that 1,187 children and adolescents were registered with T1DM across 32 health facilities in 2018. By 2023, the number had risen to over 3,000 across 35 health facilities [5]

Anecdotal data from the pediatric diabetes clinics at Mulago and Nsambya hospitals similarly suggest a notable increase in new T1DM cases between 2021–2023, with reports of new annual registrations nearly tripling. This trend is not unique to Uganda. Several countries have reported a rise in T1DM diagnoses during the COVID-19 pandemic, raising concerns about a potential link between SARS-CoV-2 infection and pancreatic beta-cell destruction [6,7]. Previous studies have implicated viruses such as enteroviruses and coxsackievirus B in the pathogenesis of T1DM, and the pandemic has renewed interest in this viral-autoimmunity hypothesis [8].

Despite these global trends, there is limited data on how the COVID-19 pandemic has influenced the onset and presentation of T1DM among children in the Sub- Saharan region. This lack of local epidemiological data limits the ability to identify patterns, plan interventions, and allocate appropriate resources for care. This study aimed to describe the trends of new-onset T1DM in children attending Mulago and Nsambya diabetes clinics before and during the COVID-19 pandemic.

## Materials and methods

### Study design

This was a retrospective cohort study based on a review of medical records of children aged 6months to 19 years newly diagnosed with T1DM during the pre-COVID-19 period between 1st March 2017 and 28th February 2020 and the COVID-19 pandemic period between 1st March 2020 and 28th February 2023.

### Study setting

The study was conducted at two major pediatric diabetes clinics in Kampala, Uganda located in Mulago National Referral hospital and St Francis hospital, Nsambya.

Mulago National Referral Hospital (MNRH) is a government funded public hospital located in the capital city of Uganda, Kampala, with a bed capacity of 1790 patients. It receives patients from all over the country and also serves as the teaching hospital for Makerere University College of Health sciences. The Paediatric diabetes clinic at MNRH is led by a Paediatric Endocrinologist supported by nurses, residents, and a medical officer. The clinic operates once a week, attending to about 20 children and adolescents per clinic day, with an average of 3 new patients per month.

St Francis Hospital, Nsambya (SFHN) is a faith based Private Not for Profit (PNFP) hospital with a capacity of 361 beds. It's also located in the capital, Kampala. The diabetes clinic is managed by a specialist Pediatrician assisted by pediatrics residents and three nursing staff. The clinic operates once a week and receives between 15–50 patients per clinic day, with an average of 2–3 new patients per month.

Both clinics are supported by the Government of Uganda and benefit from funding under World Diabetes Foundation's Changing diabetes in children (CDiC) project, which facilitates the delivery of diabetes care services to children [9]. Services at both clinics are provided free of charge, and the standard of care is consistent across the two sites.

These two paediatric diabetic clinics are referral centres for other clinics in the country however, the patients seen at these clinics are similar in characteristics to those seen elsewhere. Other children elsewhere in the country with diabetes are mainly seen at public regional hospitals in 44 facilities by clinicians including medical officers, clinical officers and some trained nurses.

### Study population

The study included medical records of all children and adolescents < 20 years who were newly diagnosed with T1DM and attended either the Mulago or Nsambya paediatric diabetes clinics over the six-year period from March 2017 to February 2023. Records with missing critical data; for example, date of diagnosis, contact information were excluded from analysis.

### Sample size

The sample size was calculated using the formula for sample size of two proportions, assuming a 95% level of confidence and 80% power. We used 12.2% and 24.4% incidence of T1DM before and during the COVID-19 pandemic respectively as reported in the worldwide sweet registry [10]. Since the population of T1DM attending the two paediatric diabetes clinic is finite, the population correction formula; n/(1+[n/N]), was employed to adjust the estimated sample size, yielding a sample size of 160 participants.

### Study variables

The dependent variable was new onset T1DM, and the independent variables were COVID 19 pandemic period (pre and during) as well as participant's clinical data notably: date of diagnosis, symptoms, duration of symptoms, family history of diabetes, history of admission for DKA, weight, length (or height), HBA1C and sociodemographic characteristics notably age, sex, residence, education level of the child, education level of parents.

## Study procedures

All eligible participant's files were included in the study. Using the T1DM registers at both study sites, all registered patients were screened to identify those newly diagnosed with T1DM.during the respective study periods. These were further reviewed for eligibility. Records were accessed and data collected concurrently from both study sites from 1st December 2023–31st March 2024.

## Statistical analysis

STATA 17 was used for the analysis. Descriptive statistics were used to summarize participant characteristics. These were compared across the two time periods using Chi square and Fisher's exact test. The p value of <0.05 was considered statistically significant. To assess trends in new onset T1DM over time, the number of new diagnoses of T1DM per month were tallied. An interrupted time series analysis was performed using 12 time points,each representing a 6 months intervals as follows; 1st March 2017–31st August 2017, 1st September 2017–28th February 2018, 1st March 2018–31st August 2018, 1st September 2018–29th February 2019, 1st March 2019–31st August 2019, 1st September 2019–29th February 2020, 1st March 2020–31st August 2020, 1st September 2020–28th February 2021, 1st March 2021–31st August 2021, 1st September 2021–28th February 2022, 1st March 2022–31st August 2022, 1st September 2022–28th February 2023. A visualization line graph was used to illustrate trends of new onset T1DM across these time points. Additionally, participant characteristics were stratified by age groups:<5 years, 5–9 years, and ≥10 years, and comparisons across the pre- and during-COVID-19 periods were conducted using Chi-squared and Fisher's exact tests, as appropriate.

## Ethical approval

Institutional approval was obtained from the Makerere University School of Medicine Research and Ethics Committee (Mak-SOMREC), approval number Mak-SOMREC-2023–666, as well as administrative clearance from Mulago National Referral Hospital and St. Francis Hospital Nsambya. A waiver of consent for data extraction from medical records was granted. However, for participants whose caregivers were contacted via telephone to obtain additional information, verbal informed consent was obtained prior to the interview.

To ensure confidentiality and privacy, each participant was assigned a unique identification number and all data were anonymized before analysis. At no point was personally identifiable information accessed or included in the study.

## Results

### Study profile

We screened a total of 400 files. Of these, 265 met the study inclusion criteria. However, 20 were excluded due to incomplete data. The study profile is illustrated in Fig 1 below.

### Characteristics of study participants

Of the 245 participants, 120 (49%) were enrolled from SFHN T1DM clinic and 125 (51%) were enrolled from the MNRH T1DM clinic. The mean age of participants was 10.9(sd 4.5) with two thirds of the participants having presented with DKA at new onset of T1DM. Only close to a quarter of those with DKA at diagnosis were managed in the HDU/ ICU. Typically, these children presented with wasting in two thirds and not obesity. The demographic and clinical characteristics are summarized in Table 1 below.

### Trends of new-onset T1DM Pre and during the COVID19 pandemic period

There was an overall increase in new-onset type 1 diabetes mellitus (T1DM) cases registered at both study clinics during the period from March 2017 to February 2023. Of the 245 total cases, 97 (39.6%) were registered before the COVID-19 pandemic (March 2017–February 2020), while 148 (60.4%) were registered during the pandemic period (March 2020–February

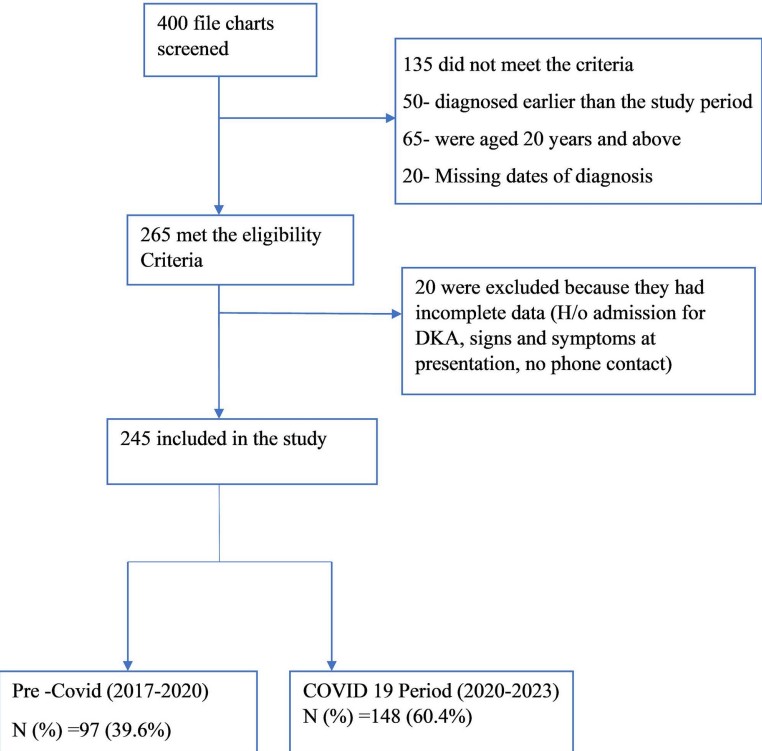

**Fig 1. Fig showing the study profile.**

2023). The trend showed cyclical variations, with the highest pre-COVID-19 peak of 22 cases recorded between March and August 2018. Following a temporary decline, cases gradually increased again leading up to the pandemic.

At the onset of the COVID-19 pandemic, new-onset T1DM cases rose to 24 between March and August 2020, followed by a slight dip and then an exponential increase peaking at 36 cases between March and August 2021. This was the highest recorded number during the study period. Although there was a subsequent drop to 18 cases by early 2022, a slow rise continued until the end of the study. On average, 16 cases were diagnosed every six months in the pre-COVID period (2.7/month), compared to 24 every six months (4/month) during the pandemic. Notably, post-COVID case numbers have not returned to pre-pandemic levels. These trends of new-onset T1DM cases are illustrated in Fig 2, while Fig 3 shows the national COVID-19 case trends for Uganda during the same period.

### Comparison of characteristics of children in the pre and COVID 19 pandemic periods

There were more children diagnosed with new onset T1DM in the COVID 19 pandemic period 148(60.4%), than in the pre-COVID period 97 (39.6%) an observed increase of 52%. The percentage of adolescents in secondary school/university that presented in the COVID 19 pandemic 26 (17.6%) was lower than that in the pre-pandemic period 34 (35.1%) (p = 0.008). Furthermore, more participants 162 of 245 (66%) presented with DKA at the time of T1DM diagnosis. The results are presented in the Table 2 below.

### Discussion

The study examined trends and characteristics of children diagnosed with T1DM at the two largest paediatric T1DM clinics in Uganda over six years, comparing the periods before and during the COVID 19 pandemic. We found a notable increase

**Table 1. Table of characteristics of the 245 participants.**

| Variable | Frequency(N) | Percentage (%) |
|---|---|---|
| **Age; mean (SD)** | 10.9 (4.5) | |
| <5 years | 29 | 11.8 |
| 5–9 years | 58 | 23.9 |
| ≥10 years | 158 | 64.6 |
| **Sex** | | |
| Female | 138 | 56.5 |
| Male | 107 | 43.5 |
| **Child's education level** | | |
| Pre-school | 41 | 16.7 |
| Primary | 144 | 58.9 |
| Secondary/ University | 60 | 24.4 |
| **Caretaker's education level** | | |
| No formal education | 63 | 25.7 |
| Primary | 33 | 13.5 |
| Secondary | 94 | 38.4 |
| University | 55 | 22.4 |
| **Location of residence** | | |
| Town/city | 74 | 30.2 |
| Trading centre | 97 | 39.6 |
| Village | 74 | 30.2 |
| **Family history of diabetes** | | |
| No | 208 | 84.9 |
| Yes | 37 | 15.1 |
| **Fever** | | |
| No | 201 | 82.0 |
| Yes | 44 | 18.0 |
| **Coma** | | |
| No | 216 | 88.1 |
| Yes | 29 | 11.9 |
| **Admitted with DKA at diagnosis** | | |
| No | 83 | 33.9 |
| Yes | 162 | 66.1 |
| **Patient admission (N = 162)** | | |
| HDU | 33 | 20.5 |
| Hospital ward | 123 | 75.8 |
| ICU | 6 | 3.7 |
| **NUTRITION STATUS** | | |
| Underweight | 161 | 65.7 |
| Normal | 73 | 29.8 |
| Overweight/ obese | 11 | 4.5 |
| **First HbA1C** median (Q1- Q3) | 13 (12 –15) | |
| **Participant Enrolment** | | |
| Nsambya clinic | 120 | 49.0 |
| Mulago clinic | 125 | 51.0 |

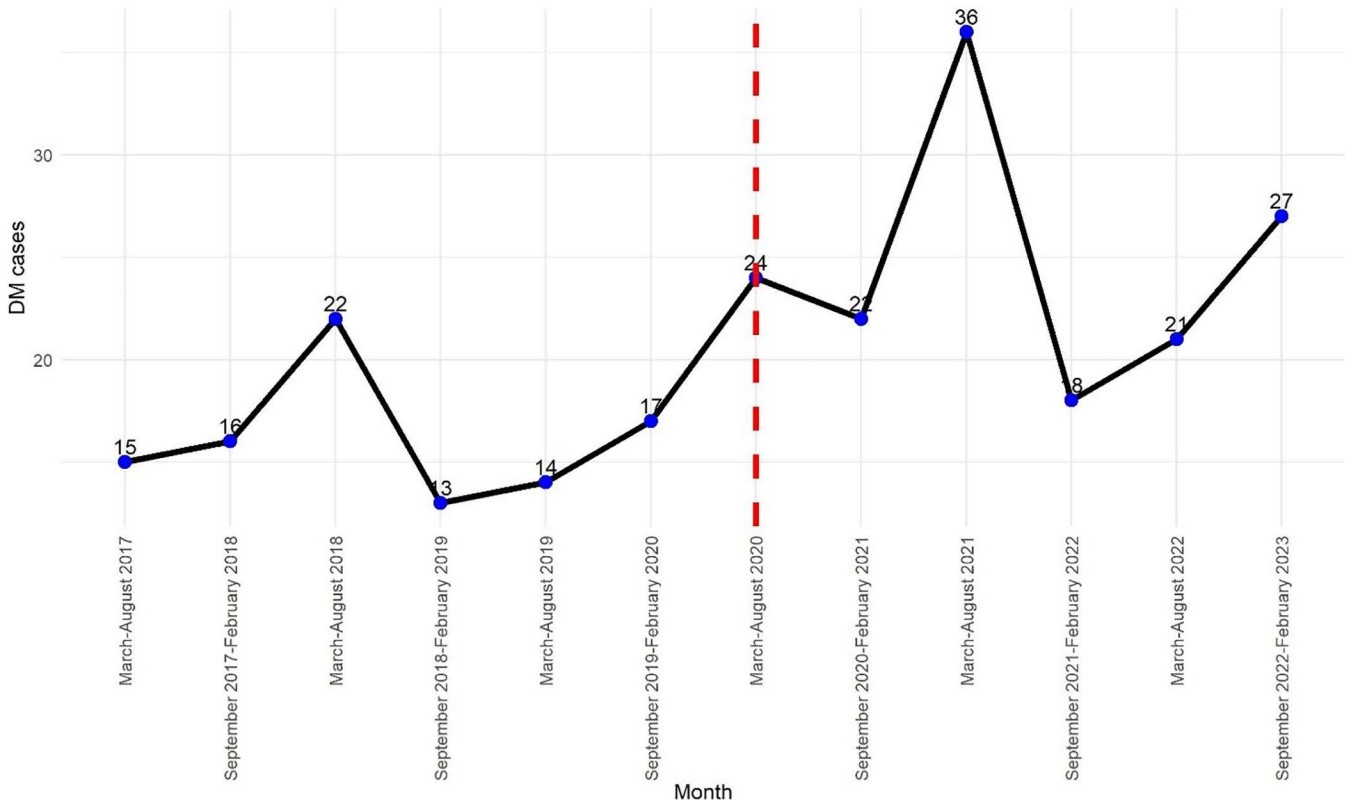

**Fig 2. Trends of New-onset T1DM in children pre and during the COVID 19 pandemic.**

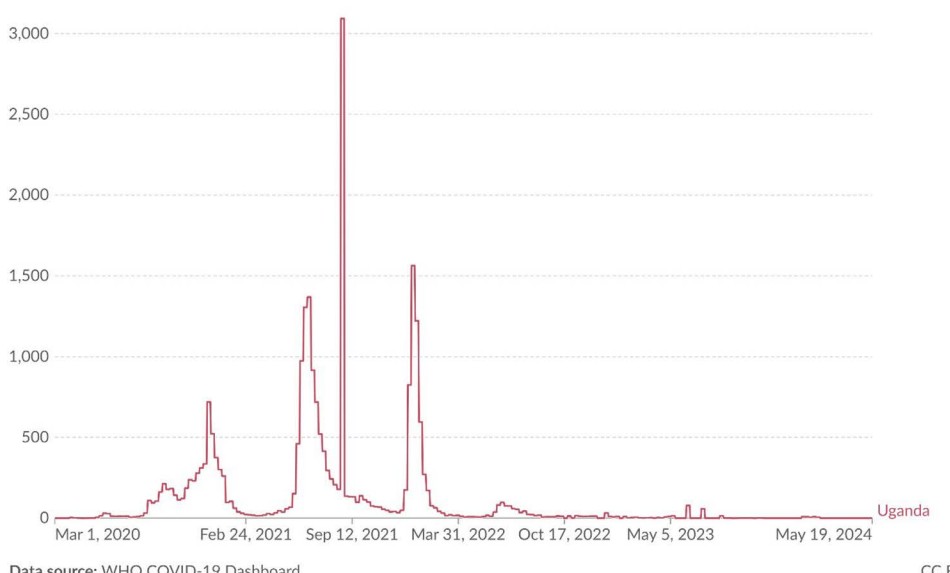

**Data source:** WHO COVID-19 Dashboard                                    CC BY

**Fig 3. Reported daily cases of SARS-COV-2 in Ugandan adults and children during the study period.**

**Table 2.** Comparison of characteristics of children in the pre and COVID 19 pandemic periods.

| Variable | Total N (%) | Pre-COVID (n = 97) Mar 2017-Feb 2020 | COVID 19 period (n = 148) Mar 2020-Feb 2023 | p-value |
|---|---|---|---|---|
| **Age** | | | | 0.187 |
| <5 years | **29 (11.8)** | **7 (7.2)** | **22 (14.9)** | |
| 5–9 years | 58 (23.9) | 25 (25.8) | 33 (22.3) | |
| ≥ 10 years | 158 (64.6) | 65 (67.0) | 93 (62.8) | |
| **Sex** | | | | 0.666 |
| Female | 138 (56.5) | 53 (54.6) | 85 (57.4) | |
| Male | 107 (43.5) | 44 (45.4) | 63 (42.6) | |
| **Child's education level** | | | | 0.008 |
| Pre-school | **41 (16.7)** | **14 (14.4)** | **27 (18.2)** | |
| Primary | 144 (58.9) | 49 (50.5) | 95 (64.2) | |
| Secondary/ University | 60 (24.4) | 34 (35.1) | 26 (17.6) | |
| **Location of residence** | | | | |
| Town/city | 74 (30.2) | 29 (29.9) | 45 (30.4) | |
| Trading centre | 97 (39.6) | 37 (38.1) | 60 (40.5) | 0.884 |
| Village | 74 (30.2) | 31 (32.0) | 43 (29.1) | 0.251 |
| **Coma** | | | | |
| No | 216 (88.2) | 88 (90.7) | 128 (86.5) | |
| Yes | 29 (11.8) | 9 (9.3) | 20 (13.5) | |
| **Family history of diabetes** | | | | 0.186 |
| No | 208 (84.9) | 86 (88.7) | 122 (82.4) | |
| Yes | 37 (15.1) | 11 (11.3) | 26 (17.6) | |
| **Admitted with DKA diagnosis** | | | | 0.167 |
| No | 83 (33.9) | 38 (39.2) | 45 (30.6) | |
| Yes | **162 (66.1)** | **60 (60.8)** | **102 (69.4)** | |
| **Patient admission (n = 162) (n = 162)** | | | | 0.361 |
| HDU | 33 (20.4) | **9 (15.0)** | **24 (23.5)** | |
| Hospital ward | 123 (75.9) | 49 (81.7) | 74 (72.5) | |
| ICU | 6 (3.7) | **2 (3.3)** | **4 (4.0)** | |
| **Nutrition Status** | | | | 0.909 |
| Underweight | 161 (65.7) | 63 (64.9) | 98 (66.2) | |
| Normal weight | 73 (29.8) | 29 (29.9) | 44 (29.7) | |
| Overweight/Obese | 11 (4.5) | 5 (5.2) | 6 (4.1) | |
| First **HbA1C**, mean (SD) | | 16.1 (17.3) | 13.3 (3.05) | 0.151 |
| **Duration of coma; Med(IQR) (Q1QQQ3)** | 29 | 1 (1–2) | 1 (1–1) | 0.35 |

in new onset T1DM during the pandemic with a 52% rise, and a doubling in monthly cases, although it was not statistically significant (p = 0.1997). The increase in incident cases of T1DM was reported elsewhere [11–15].

The peak in T1DM cases occurred between highest increase March to August 2021, coinciding with Uganda's second COVID-19 wave that was dominated by the delta variant. Uganda also reported that highest number of COVID-19 cases during this same period [16,17]. Although children are often asymptomatic carriers of SARS CoV-2, indirect exposure from high community transmission likely contributed to increase in cases among children and these cases may not have been recorded by the ministry of health. This observation underscores the hypothesis that SARS CoV- 2 has been implicated in direct beta cell damage, increased autoimmune damage and potentially triggering T1DM [18,19]. Despite earlier

assumptions that lockdowns delayed access to care [7], our data showed no significant difference in symptom duration or severity of disease (DKA) presentation between the two periods, suggesting that the increase was more likely due to SARS-CoV 2 exposure. Interestingly, more children<5 years were diagnosed during the pandemic, suggesting a shift in age distribution in our population from higher mean age of diagnosis of 12 years [5] to lower mean age of diagnosis of 10 years possibly due to increased supervision during the lockdown. This aligns well with global observations of more acute childhood T1DM presentations during the pandemic [10]

More children presented in DKA coma during the COVID 19 period although did not reach statistical significance, suggesting a more severe disease due to plausibly delay in accessing care due to lock down measures. This was reported elsewhere by Yang and colleagues who reported that SARS-CoV 2 was associated with more aggressive T1DM course and easier DKA development [18]. In China, Li and colleagues observed that COVID 19 induced ketosis and ketoacidosis in some patients [19]. In India [20] and in a region of US [21], the proportion of patients with DKA requiring PICU admission increased in the COVID pandemic period. More other studies in different countries unanimously agree that there was increased incidence of new onset T1DM and DKA in the COVID 19 period [22,23].

Globally, the WHO attributes the fivefold increase in Childhood T1DM to among other things increase in childhood obesity [24]. Although some studies documented an increase in childhood obesity in the COVID 19 pandemic [25], in our study most children in both periods were underweight, implying that this was most likely T1DM and not T2DM. However, there has also been reports of increased T1DM due to the increase in childhood type 1 diabetes during the COVID-19 pandemic may be due to autoimmunity [26].

This study was in Uganda's largest pediatric diabetes clinics and this regional data contributes to the global conversation on T1DM and viral infections particularly COVID-19.

However, Children were not screened for COVID-19 antibodies, limiting the ability to establish causality and the retrospective design posed challenges with missing data and potential recall bias despite efforts to contact caregivers for additional information.

In conclusion, there was a rising trend in T1DM among Ugandan children at two largest pediatric diabetic clinics in Kampala, at Mulago and Nsambya hospitals, from 2.7 new cases/month pre-COVID to 4.1/month during the pandemic. Two-thirds of the children presented with DKA at diagnosis, indicating low awareness of early T1DM symptoms and the need for improved recognition.

We recommend prospective studies to assess the causal link between SARS-CoV-2 and T1DM. National-scale studies are needed to accurately estimate the incidence of T1DM in Ugandan children. There is also need for increased health community awareness of T1DM to promote early diagnosis and reduce complications.

## Supporting information

**S1 File. Data**.
(XLS)

## Author contributions

**Conceptualization:** Rosemary Ahabwenki, Thereza Piloya, Flavia Namiiro.

**Data curation:** Rosemary Ahabwenki.

**Formal analysis:** Wani Muzeyi, Thomas Katairo.

**Funding acquisition:** Sarah Kiguli.

**Investigation:** Rosemary Ahabwenki, Thereza Piloya, Flavia Namiiro, Catherine Nyangabyaki.

**Methodology:** Rosemary Ahabwenki, Wani Muzeyi.

**Project administration:** Rosemary Ahabwenki.

**Resources:** Rosemary Ahabwenki, Sarah Kiguli.

**Software:** Wani Muzeyi, Thomas Katairo.

**Supervision:** Thereza Piloya, Flavia Namiiro, Catherine Nyangabyaki, Bernard Kikaire, Victor Musiime.

**Validation:** Rosemary Ahabwenki, Flavia Namiiro.

**Visualization:** Rosemary Ahabwenki, Thereza Piloya.

**Writing – original draft:** Rosemary Ahabwenki, Thereza Piloya.

**Writing – review & editing:** Rosemary Ahabwenki, Thereza Piloya, Flavia Namiiro, Wani Muzeyi, Catherine Nyangabyaki, Aidah Namugumya, Joan Nakimera, Fred Kyekulidde, Robert Kidega, Fozia Nizar Ali, Angella Mirembe, Bernard Kikaire, Grace Ndeezi, Victor Musiime, Sarah Kiguli.

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
