## [Decision Letter · Decision Letter 0]

5 Aug 2025

Dear Dr. AHABWENKI,

Thank you for submitting your manuscript to PLOS ONE. After careful consideration, we feel that it has merit but does not fully meet PLOS ONE’s publication criteria as it currently stands. Therefore, we invite you to submit a revised version of the manuscript that addresses the points raised during the review process.

We look forward to receiving your revised manuscript.

Kind regards,

Masoud Rahmati

Academic Editor

PLOS ONE

Journal Requirements:

[This work was supported by the Fogarty International Center of the National Institutes of Health, U.S Department of State’s Office of the U.S Global AIDS Coordinator and Health Diplomacy (S/GAC), and Presidents Emergency Plan for AIDS’ Relief (PEPFAR) under award number 1R25TW011213. The content is solely the responsibility of the authors and does not necessarily represent the official views of the National Institutes of Health.].

Reviewers' comments:

Reviewer's Responses to Questions

**Comments to the Author**

1. Is the manuscript technically sound, and do the data support the conclusions?

Reviewer #1: No

2. Has the statistical analysis been performed appropriately and rigorously?

Reviewer #1: No

3. Have the authors made all data underlying the findings in their manuscript fully available?

Reviewer #1: Yes

4. Is the manuscript presented in an intelligible fashion and written in standard English?

Reviewer #1: Yes

Reviewer #1: This is interesting data and it comes from the part of the world which does not enjoy too much attention. This is a study of changes in new onset DM in two clinics in Kampala Uganda and the possible increase in DM 1 incidence since the start of the COVID pandemic. The issue for me is that the conclusions that there's increased incidence in the whole country are not supported by the data which is rather modest and the findings are not statistically significant (per figure 2). Quick googling tells me that Uganda has about 22 million children and assuming a pretty conservative prevalence of DM of 0.15% (half of US prevalence) there should be around 38 thousand children with DM1 in Uganda and the yearly incidence should be just under 2 thousand. This study has only 400 charts of whom 265 were new onset. Expanding the findings of this small subset to the entire country is overly ambitious. Additionally figure 2 shows that the pre and post covid difference in number of cases per half year period did not reach statistical significance and looking at the graph, the increase seems to be mostly driven by the COVID peak. So overall, I think that the findings of this paper need to be rethought.

Details:

Introduction:

1. there are many more papers that need to be cited ( for example https://doi.org/10.1002/jmv.27996, https://doi.org/10.2337/dc21-0969, https://doi.org/10.1155/2023/4580809, https://doi.org/10.1016/j.dsx.2021.02.009)

2. an explanation is needed for how these two clinics are meant to be stand ins for the entrire country.

Methods:

1 until what age are these patients seen/included in the study?

2. there's lots of unnecessary details about how the clinics are run but no explanation for where else patients might be going. Is there selection bias in who comes to the clinics?

Results:

1. main finding of the paper is not statistically significant, ie is not really a finding

Discussion:

Paragraph one is not entirely true in view of above

**Do you want your identity to be public for this peer review?** For information about this choice, including consent withdrawal, please see our Privacy Policy

Reviewer #1: No

---

## [Author Response · Author response to Decision Letter 1]

24 Sep 2025

All comments made by the reviewers and editors have been responded to. The responses are in the uploaded document entitled Response to Reviewers.

---

## [Decision Letter · Decision Letter 1]

29 Oct 2025

Dear Dr. AHABWENKI,

Thank you for submitting your manuscript to PLOS ONE. After careful consideration, we feel that it has merit but does not fully meet PLOS ONE’s publication criteria as it currently stands. Therefore, we invite you to submit a revised version of the manuscript that addresses the points raised during the review process.

We look forward to receiving your revised manuscript.

Kind regards,

Masoud Rahmati

Academic Editor

PLOS ONE

Journal Requirements:

Reviewers' comments:

Reviewer's Responses to Questions

**Comments to the Author**

Reviewer #1: (No Response)

2. Is the manuscript technically sound, and do the data support the conclusions?

Reviewer #1: No

3. Has the statistical analysis been performed appropriately and rigorously?

Reviewer #1: Yes

4. Have the authors made all data underlying the findings in their manuscript fully available?

Reviewer #1: Yes

5. Is the manuscript presented in an intelligible fashion and written in standard English?

Reviewer #1: Yes

Reviewer #1: Response to response:

Introduction:

1. there are many more papers that need to be cited ( for example https://doi.org/10.1002/jmv.27996, https://doi.org/10.2337/dc21-0969,
https://doi.org/10.1155/2023/4580809, https://doi.org/10.1016/j.dsx.2021.02.009)

Response: These have been cited in the last sentence of the first paragraph of the discussion section. References 12-15.

New Response: great

2. an explanation is needed for how these two clinics are meant to be stand ins for the entire country.

Response: These two pediatric diabetic clinics at Mulago and Nsambya hospitals are the largest main treatment and referral centres for childhood diabetes in Uganda. They treat the majority of Ugandan pediatric T1DM and are referral centres for T1DM for the whole country. The statement on findings and conclusions has been clarified that the findings of this study were at the two clinics in Kampala, Uganda.

New Response: you do not address my main commentary. Until this is addressed, I cannot allow this paper to proceed. How can these very few patients be a stand in for the whole country? There should be thousands of kids of DM and you access a few hundred. If my math in the previous review was incorrect, please explain to me why. You don't have to do it in the paper itself if my argument makes no sense, but you have to address it.

Methods:

1 until what age are these patients seen/included in the study?

Response: From ages 6months to 19years. This was already mentioned in the abstract in the methods section, and has been re emphasised in the first paragraph of the study design under methods section.

New Response: great

2. there's lots of unnecessary details about how the clinics are run but no explanation for where else patients might be going. Is there selection bias in who comes to the clinics?

Response: There is no difference in the sociodemographic characteristics of children with T1DM who are seen at either Mulago or Nsambya diabetic clinics. Services provided are similar at both clinics and free of charge under the support of the Government of Uganda and World Diabetes Foundation’s Changing diabetes in children (CDiC) project. A few children with T1DM seen at Lubaga Hospital, and others diagnosed from other hospitals in the country are linked to these two clinics for treatment support and follow up.

New Response: My question was not about the difference between the two clinics but between patients who are seen in these clinics and those who are not. Harking back to my main point, thousands of Ugandan kids are unaccounted for here. Who sees them? Are they different? How?

Results:

1. main finding of the paper is not statistically significant, ie is not really a finding

Response: This has been clarified that it was not statistically significant

New Response: Great

Discussion:

Paragraph one is not entirely true in view of above

Response: This has been clarified that much as there was an observed increase in the trend of T1DM, it was not statistically significant.

New Response: Great

**Do you want your identity to be public for this peer review?** For information about this choice, including consent withdrawal, please see our Privacy Policy

Reviewer #1: Yes:  Renee Bargman

---

## [Author Response · Author response to Decision Letter 2]

17 Dec 2025

All comments raised by the reviewer have been responded to, and documents attached

All personal information has been removed from the data sets, and only de identified data has been shared

---

## [Decision Letter · Decision Letter 2]

4 Jan 2026

Impact of COVID-19 on New – onset Type 1 diabetes mellitus: A six-year retrospective review from two Paediatric clinics, Kampala, Uganda.

PONE-D-25-31814R2

Dear Dr. AHABWENKI,

We’re pleased to inform you that your manuscript has been judged scientifically suitable for publication and will be formally accepted for publication once it meets all outstanding technical requirements.

Kind regards,

Masoud Rahmati

Academic Editor

PLOS One

Additional Editor Comments (optional):

Reviewers' comments:

Reviewer's Responses to Questions

**Comments to the Author**

Reviewer #1: All comments have been addressed

2. Is the manuscript technically sound, and do the data support the conclusions?

Reviewer #1: Yes

3. Has the statistical analysis been performed appropriately and rigorously?

Reviewer #1: Yes

4. Have the authors made all data underlying the findings in their manuscript fully available?

Reviewer #1: Yes

5. Is the manuscript presented in an intelligible fashion and written in standard English?

Reviewer #1: Yes

Reviewer #1: comments have been addressed, ok to proceed with publication. the authors addressed my concerns. why do I need more characters here?

**Do you want your identity to be public for this peer review?** For information about this choice, including consent withdrawal, please see our Privacy Policy

Reviewer #1: Yes:  Renee Bargman

---

## [Editor Report · Acceptance letter]

PONE-D-25-31814R2

PLOS One

Dear Dr. AHABWENKI,

I'm pleased to inform you that your manuscript has been deemed suitable for publication in PLOS One. Congratulations! Your manuscript is now being handed over to our production team.

Kind regards,

on behalf of

Dr. Masoud Rahmati

Academic Editor

PLOS One